# QGOpt: Riemannian optimization for quantum technologies

I. A. Luchnikov[1,2], A. Ryzhov[2], S. N. Fillipov[1,3,4] H. Ouerdane[2*]

**1** Moscow Institute of Physics and Technology, Institutskii Pereulok 9, Dolgoprudny, Moscow Region 141700, Russia
**2** Center for Energy Science and Technology, Skolkovo Institute of Science and Technology, Moscow 121205, Russia
**3** Steklov Mathematical Institute of Russian Academy of Sciences, Gubkina Street 8, Moscow 119991, Russia
**4** Valiev Institute of Physics and Technology of Russian Academy of Sciences, Nakhimovskii Prospect 34, Moscow 117218, Russia
* h.ouerdane@skoltech.ru

November 7, 2020

## Abstract

**Many theoretical problems in quantum technology can be formulated and addressed as constrained optimization problems. The most common quantum mechanical constraints such as, e.g., orthogonality of isometric and unitary matrices, CPTP property of quantum channels, and conditions on density matrices, can be seen as quotient or embedded Riemannian manifolds. This allows to use Riemannian optimization techniques for solving quantum-mechanical constrained optimization problems. In the present work, we introduce QGOpt, the library for constrained optimization in quantum technology. QGOpt relies on the underlying Riemannian structure of quantum-mechanical constraints and permits application of standard gradient based optimization methods while preserving quantum mechanical constraints. Moreover, QGOpt is written on top of TensorFlow, which enables automatic differentiation to calculate necessary gradients for optimization. We show two application examples: quantum gate decomposition and quantum tomography.**

# 1 Introduction

Application of the variational method in quantum physics is akin to a constrained optimization problem. For example, the ground state of a quantum system with Hamiltonian $H$ can be found by use of the variational principle [1]:

$$|\Omega\rangle = \operatorname*{argmin}_{|\psi\rangle} \frac{\langle\psi|\,H\,|\psi\rangle}{\langle\psi|\psi\rangle}, \tag{1}$$

where $|\psi\rangle$ is a trial, non-normalized state, $|\Omega\rangle$ is the non-normalized ground state. This formulation of a ground state search problem was successfully used for characterization of many-body quantum systems [2, 3]. In particular, the ground state of a correlated spin system can be found in the following forms: matrix product states [4–6], projected entangled pair states [7, 8] or neural networks [9–11]. To perform optimization of variational energy one can utilize optimization algorithms such as density matrix renormalization group [12, 13], time evolving block decimation [14–16] for tensor network architectures, quantum natural gradient [17], and adaptive first order optimization methods like Adam optimizer [18] for neural networks based quantum parametrization.

Problems of reconstruction of quantum states, quantum channels, quantum processes, etc. from measurements data can also be formulated as optimization problems. For example, the state of a many-body quantum system can be reconstructed in a neural network form [19–22] by maximization of the logarithmic likelihood function on a set of measurement outcomes. The Choi matrix of an unknown quantum channel can be reconstructed in a form of tensor network via Kullback-Leibler divergence minimization [23]. Model of non-Markovian quantum dynamics can be reconstructed from measured data in different ways [24, 25] by use of optimization algorithms.

Some problems of quantum mechanics require nonstandard optimization methods. For example, a well known entanglement renormalization technique, that is used for characterization of quantum phase transitions, requires to perform optimization over isometric matrices. To solve this problem, Vidal and Evenbly suggested an algorithm [26–28] that does not have analogs in standard optimization theory. Further, some optimization problems such as entanglement renormalization or quantum tomography, require preservation of natural "quantum" constraints, such as the completely positive and trace preserving (CPTP) property of quantum channels [29] or the orthogonality constraints of isometric or unitary matrices. Preservation of constraints can be achieved by introduction of a particular parametrization or by adding a regularization term to a loss function that enforces the satisfaction of constraints. However, a naive parametrization may lead to over-parametrization, which causes optimization slowing down.

Highly specialized algorithms such as the Vidal–Evenbly algorithm, are only suitable for a restricted set of problems. Additional regularization terms in a loss function are also not a universal solution because they provide only approximate constraints preservation. One therefore needs a universal approach to the optimization in quantum technology. As many natural "quantum" constraints can be seen as Riemannian manifolds, Riemannian

optimization is a right candidate for the role of a universal framework for constrained optimization in quantum mechanics. In the present work, we introduce QGOpt (Quantum Geometric Optimization) [30], our library for Riemannian optimization in quantum mechanics and quantum technologies. It allows one to perform an optimization with all typical constraints of quantum mechanics.

This article is organized as follows. In Sec. 2, we give an overview of Riemannian optimization. We then turn to Riemannian manifolds in quantum mechanics in Sec. 3. In Sec. 4, we present the QGOpt application programming interface (API), and we illustrate its utilization in Sec. 5, with two examples: quantum gate decomposition and quantum channel tomography.

## 2 Overview of the Riemannian optimization

While optimizing an objective function defined on the Euclidean space, one performs a sequence of elementary operations like points and vectors transportation. Optimization on curved spaces requires a generalization of these elementary operations in a certain way. As an example we consider a case of gradient descent with momentum [31] and its Riemannian generalization [32, 33]. Here, we keep our overview simple. For an in-depth introduction into the topic, we recommend the following references [34, 35].

Let us assume that we aim to minimize the value of a function $f : \mathbb{R}^n \longmapsto \mathbb{R}$, and that we have access to its gradient $\nabla f(x)$. In the Euclidean space $\mathbb{R}^n$, a gradient descent with momentum consists of the following steps wrapped into a loop:

1. Calculating the momentum vector $m_{t+1} = \beta m_t + (1 - \beta)\nabla f(x_t)$,

2. Taking a step along the direction of a momentum vector $x_{t+1} = x_t - \eta m_{t+1}$,

where the initial momentum vector $m_0$ is the null vector, $\beta$ is a hyperparameter whose value is usually taken around $\beta \approx 0.9$, and $\eta$ is the size of the optimization step. The sign before $\eta$ indicates whether we search for a local minimum or maximum.

Let us assume now that a function $f$ is defined on a Riemannian manifold $\mathcal{M}$ that is embedded in the Euclidean space: $f : \mathcal{M} \longmapsto \mathbb{R}$. Then we can no longer apply the standard scheme of gradient descent with momentum, because it clearly takes $x_t$ out of the manifold $\mathcal{M}$. This scheme can be generalized step by step. First, we have to generalize the notion of a gradient. The standard Euclidean gradient is not a tangent vector to a manifold and it does not take into account the metric of a manifold. One may then introduce the Riemannian gradient that can be constructed based on the standard gradient $\nabla f(x)$. The Riemannian gradient lies in the space tangent to a point $x$ and properly takes the metric of a tangent space into account. Although an optimization algorithm takes a step along a tangent vector to a manifold, it still takes a point out of the manifold. In order to fix this issue, one can replace a straight line step with a curved line step that is called retraction. The retraction $R_x(v)$ is a result of the transportation of $x$ along a curve that completely lies in the manifold and directed along $v$.

The gradient descent with momentum also requires to transport the momentum vector at each iteration from a previous point to a new point. The Euclidean version of the gradient descent with momentum does not have an explicit step with transportation of the momentum vector, because in the Euclidean space transportation of a vector is trivial. However, this step is necessary in the Riemannian case, where the trivial Euclidean vector transportation takes a vector out of a tangent space. A vector transport $\tau_{x,w}(v)$ is the result of transportation of a vector $v$ along a vector $w$ which takes into account that a

tangent space varies from one manifold's point to another in the Riemannian case. The overall Riemannian generalization of the gradient descent with momentum reads:

1. Calculating the momentum vector $\tilde{m}_{t+1} = \beta m_t + (1 - \beta) \nabla_R f(x_t)$,

2. Taking a step along a new direction of the momentum $x_{t+1} = R_{x_t}(-\eta \tilde{m}_{t+1})$,

3. Transporting of the momentum vector to a new point $x_{t+1}$: $m_{t+1} = \tau_{x_t, -\eta \tilde{m}_{t+1}}(\tilde{m}_{t+1})$.

Other first-order optimization methods can be generalized in a similar fashion.

## 3  Riemannian manifolds in quantum mechanics

Many objects of quantum mechanics can be seen as elements of smooth manifolds. However, their mathematical description, suitable for numerical algorithms, may involve some abstract constructions that should be clarified. In this section we consider an illustrative example of a Choi matrices set and describe this set as a smooth quotient manifold. We restrict our consideration to a plain description of all necessary mathematical concepts. At the end of the section, we also list all the manifolds implemented in the QGOpt library and describe their possible use.

The evolution of any quantum system that interacts with its environment can be described by a quantum channel. Here, we consider quantum channels defined as the following CPTP linear map: $\Phi : \mathbb{C}^{n \times n} \longmapsto \mathbb{C}^{n \times n}$. Any quantum channel can be represented through its Choi matrix [29]. A Choi matrix is a positive semi-definite operator $C \in \mathbb{C}^{n^2 \times n^2}$ that has a constraint $\mathrm{Tr}_p(C) = \mathbb{1}$, where $\mathrm{Tr}_p$ is a partial trace over the first subsystem and $\mathbb{1}$ is the identity matrix. To make the notion of the partial trace less abstract, let us consider a piece of TensorFlow code, that computes a partial trace of a Choi matrix. First of all, we apply a reshape operation to a Choi matrix that changes the shape of a matrix as follows

```
1  C_resh = tf.reshape(C, (n, n, n, n)).
```

The tensor $C_{\mathrm{resh}} \in \mathbb{C}^{n \times n \times n \times n}$ is an alternative representation of the Choi matrix. Further in the text, we distinguish two equivalent representations of a Choi matrix: $C$ and $C_{\mathrm{resh}}$. Partial trace of a Choi matrix can be calculated by using $C_{\mathrm{resh}}$ as follows $[\mathrm{Tr}_p(C)]_{i_1 i_2} = \sum_j [C_{\mathrm{resh}}]_{i_1 j i_2 j}$. Practically it can be done by running of the following line of code:

```
1  tf.einsum('ikjk->ij', C_resh),
```

which means that we take a trace over first and third indices (numeration of indices starts from 0).

The Choi–Jamiołkowski isomorphism [36] establishes a one-to-one correspondence between quantum channels and Choi matrices. One can calculate the Choi matrix of a known quantum channel as follows

$$C = \mathbb{1} \otimes \Phi \, |\Psi^+\rangle \langle \Psi^+| , \tag{2}$$

where $|\Psi^+\rangle = \sum_{i=1}^{n} |i\rangle \otimes |i\rangle$ and $\{|i\rangle\}_{i=1}^{n}$ is an orthonormal basis in $\mathbb{C}^n$. In order to show that the Choi matrix essentially is a quantum channel itself, we consider the representation of Eq.(2) in terms of tensor diagrams [37, 38]. The reshaped version of a Choi matrix $[C_{\mathrm{resh}}]_{i_1 j_1 i_2 j_2}$ is shown in Fig. 1. Thus, one can conclude that the Choi matrix is a quantum channel itself.

One can see that the set of all Choi matrices of size $n^2 \times n^2$ (the corresponding quantum channel acts on density matrices of size $n \times n$) $C_n$ is the following subset of $\mathbb{C}^{n^2 \times n^2}$

$$C_n = \left\{ C \in \mathbb{C}^{n^2 \times n^2} \big| C \geq 0, \ \mathrm{Tr}_p(C) = \mathbb{1} \right\}, \tag{3}$$

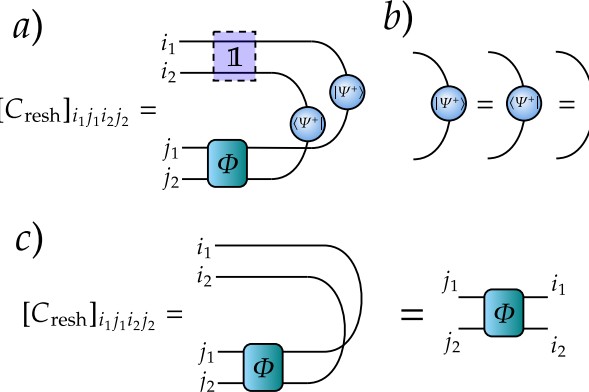

Figure 1: a) Diagrammatic representation of the Choi matrix. b) One can note that the state of a two-component quantum system $|\Psi^+\rangle$ can be seen as the identity matrix. c) Finally, we note that the Choi matrix is a quantum channel itself.

a)
$$[C_{\text{resh}}]_{i_1 j_1 i_2 j_2} = \quad \boxed{A^*} - \boxed{A}$$

b)
$$\sum_j [C_{\text{resh}}]_{i_1 j i_2 j} = \quad \boxed{A^*} - \boxed{A} \quad = \quad {}^{i_1}\!\!-\!\!{}^{i_2} = \mathbb{1}$$

Figure 2: a) Decomposition of a Choi matrix into $A$ and $A^\dagger$. b) Diagrammatic representation of the isometric property of $A$.

where $C \geq 0$, $\text{Tr}_p(C) = \mathbb{1}$ corresponds to the CPTP property of the corresponding quantum channel. This subset can be described as a Riemannian manifold that admits different Riemannian optimization algorithms. Now, we may parametrize the Choi matrix with an auxiliary matrix $A \in \mathbb{C}^{n^2 \times n^2}$

$$C = A^\dagger A. \tag{4}$$

The matrix $C$ is a positive semi-definite by construction. We also distinguish $A \in \mathbb{C}^{n^2 \times n^2}$ and its reshaped version $A_{\text{resh}} \in \mathbb{C}^{n^2 \times n \times n}$ that are connected by the reshape operation. The condition on a partial trace of a Choi matrix transforms to the following equality:

$$[\text{Tr}_p(C)]_{i_1 i_2} = [\text{Tr}_p(A^\dagger A)]_{i_1 i_2} = \sum_{kj} [A_{\text{resh}}]^*_{k i_1 j} [A_{\text{resh}}]_{k i_2 j} = \delta_{i_1 i_2} \tag{5}$$

which shows that a set of all $A_{\text{resh}}$ is the set of isometric tensors. Indeed, one can notice that convolution between $A_{\text{resh}}$ and its complex conjugated version $A^*_{\text{resh}}$ is the identity matrix which means that any $A_{\text{resh}}$ is an isometric tensor. A diagrammatic form of the Eq.(5) is shown in Fig. 2. The set of all complex isometric matrices of fixed size forms a Riemannian manifold called complex Stiefel manifold [39] that we denote as St.

At first glance, it looks like we have shown that the set of Choi matrices can be seen as a Stiefel manifold, but there is a problem that invalidates this statement: the matrices $A$ and $QA$, where $Q$ is an arbitrary unitary matrix, correspond to the same Choi matrix; in other words we have an equivalence relation $QA \sim A$. Indeed

$$C = A^\dagger Q^\dagger Q A = A^\dagger A. \tag{6}$$

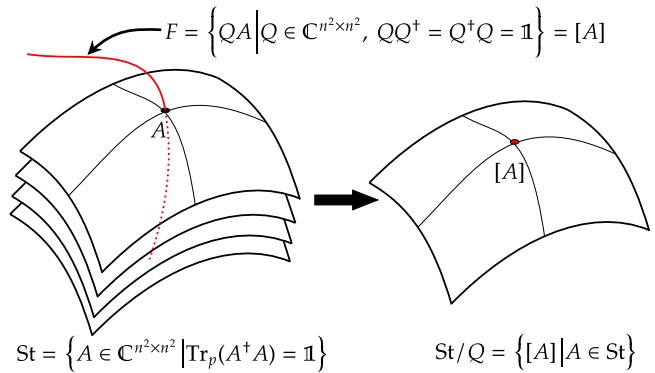

Figure 3: Diagrammatic representation of the Eq.(6.)

$$F = \left\{ QA \,\Big|\, Q \in \mathbb{C}^{n^2 \times n^2},\ QQ^\dagger = Q^\dagger Q = \mathbb{1} \right\} = [A]$$

$$\mathrm{St} = \left\{ A \in \mathbb{C}^{n^2 \times n^2} \,\Big|\, \mathrm{Tr}_p(A^\dagger A) = \mathbb{1} \right\} \qquad \mathrm{St}/Q = \left\{ [A] \,\Big|\, A \in \mathrm{St} \right\}$$

Figure 4: Graphical representation of the transition from the manifold St of all matrices $A$, to the quotient manifold St$/Q$ that eliminates undesirable symmetry of the parametrization. Red curve represents a particular equivalence class $F$ that is also called a fiber.

A diagrammatic version of Eq.(6) is depicted in Fig. 3. It shows that for any $A$ there is a family of equivalent matrices $[A] = \{QA | Q \in \mathbb{C}^{n^2 \times n^2},\ Q^\dagger Q = QQ^\dagger = \mathbb{1}\}$ that is called equivalence class of $A$, and leads to the same Choi matrix. One can eliminate this symmetry by turning to a quotient manifold St$/Q = \{[A] | A \in \mathrm{St}\}$ that consists of equivalence classes. This rather abstract construction can be imagined as a projection of a manifold along surfaces representing equivalence classes (see Fig. 4). Having a map $\pi(A) = [A]$ and a map called horizontal lift [34], that connects tangent spaces of St$/Q$ and tangent spaces of St, one can describe the abstract manifold St$/Q$ through St. It allows one to perform a Riemannian optimization on the abstract quotient manifold St$/Q$.

The example Choi matrices manifold shows all the necessary steps that emerge while building the mathematical description of quantum mechanical manifolds. The set of all manifolds implemented in QGOpt library is listed below.

- Complex Stiefel manifold $\mathrm{St}_{n,p} = \left\{ V \in \mathbb{C}^{n \times p} | V^\dagger V = \mathbb{1} \right\}$ is a set of all isometric matrices of fixed size. A particular case of this manifold is a set of all unitary matrices of fixed size; therefore, this manifold can be used for different tasks related to quantum control. Some architectures of tensor networks may include isometric matrices as building blocks [40, 41]; thus, one can use this manifold to optimize such tensor networks.

- Quotient manifold of density matrices $\varrho_n = \left\{ \varrho \in \mathbb{C}^{n \times n} \,\Big|\, \varrho = \varrho^\dagger,\ \mathrm{Tr}(\varrho) = 1,\ \varrho \succeq 0 \right\}$ is a set of all fixed-size Hermitian positive semi-definite matrices with unit trace. Since density matrices represent states of quantum systems, one can use this manifold to perform state tomography and optimization of initial quantum states in different quantum circuits.

- Quotient manifold of Choi matrices $C_n = \left\{ C \in \mathbb{C}^{n^2 \times n^2} \,\Big|\, C = C^\dagger,\ \mathrm{Tr}_p(C) = \mathbb{1},\ C \succeq 0 \right\}$ is a set of all fixed-size hermitian positive semi-definite matrices with auxiliary lin-

ear constraint (equality of the partial trace to the identity matrix). Choi matrices are used as representations of quantum channels; hence, one may use this manifold to perform quantum channel tomography and optimization of quantum channels in different quantum circuits.

- Manifold of Hermitian matrices $H_n = \left\{ H \in \mathbb{C}^{n \times n} \middle| H = H^\dagger \right\}$ is essentially a linear subspace of a space $\mathbb{C}^{n \times n}$. Since Hermitian matrices represent measurable physical operators in the quantum theory, one can use this manifold to perform a search of optimal measurable physical operators in different problems.

- Manifold of hermitian positive definite matrices $\mathbb{S}_{++}^n = \left\{ S \in \mathbb{C}^{n \times n} \middle| S = S^\dagger, \ S \succ 0 \right\}$ is a set of all positive definite matrices of fixed size. One can use it to search the optimal non-normalized quantum state in different tasks.

- Quotient manifold of positive operator-valued measure (POVM) $\text{POVM}_{m,n} = \left\{ \{E_i\}_{i=1}^n \in \mathbb{C}^{n \times d \times d} \middle| E_i = E_i^\dagger, \ E_i \succeq 0, \ \sum_{i=1}^n E_i = \mathbb{1} \right\}$ can be considered as a fixed-size tensor with hermitian positive semi-definite slices that sum into the identity matrix. Since POVMs describe generalized measurements in quantum theory, one can use this manifold to perform a search of optimal measurements that give the largest information gain.

## 4 QGOpt API

### 4.1 Manifolds API

The central class of the QGOpt library is the manifold base class. All particular manifold types are inherited from the manifold base class. All manifold subclasses admit working with the direct product of several manifolds. Any manifold has a set of typical methods that are used in Riemannian optimization. This list of methods allows one not to pay particular attention to the underlying Riemannian geometry details.

Let us consider basic illustrative examples. First, one needs to import all necessary libraries and create an example of a manifold. As an example we consider the complex Stiefel manifold.

```
1 import QGOpt as qgo
2 import tensorflow as tf
3
4 # example of complex Stiefel manifold
5 m = qgo.manifolds.StiefelManifold()
```

Here $m$ is an example of the complex Stiefel manifold that contains all the necessary information on the manifold's geometry. Some manifolds allow one to specify a type of metric and retraction as well. Using this example of a manifold one can sample a random point from a manifold:

```
1 u = m.random((4, 3, 2))
```

Here we sample a random tensor $u$, that is a complex valued TensorFlow tensor of size $4 \times 3 \times 2$. This tensor represents a point from the direct product of four complex Stiefel manifolds. The first index of this tensor enumerates a manifold and the last two indices are matrix indices. Therefore, the tensor $u$ can be seen as a set of four isometric matrices. One can generate a random tangent vector drawn from $u$.

```
1 v = m.random_tangent(u)
```

Here $v$ is a complex valued TensorFlow tensor of the same size and type as $u$ that represents the random tangent vector drawn from $u$. Now let us assume that we have a random vector $w$ which is of the same size and type, but is not tangent. One can make the orthogonal projection of this vector on the tangent space of $u$:

```
1 w = m.proj(u, w)
```

The updated vector $w$ is an element of the tangent space of $u$ now. To get the scalar product of two tangent vectors one can use the following line of code:

```
1 wv_inner = m.inner(u, w, v)
```

Here we pass $u$ to the inner product method to specify the tangent space where we compute the inner product, because in Riemannian geometry the metric and inner product are point-dependent in general.

To implement first-order Riemannian optimization methods on a manifold one needs to be able to move points and vectors along the manifold. There are retraction and vector transport methods for this purpose. As an example let us move a point $u$ along a tangent vector $v$ via the retraction map:

```
1 u_tilde = m.retraction(u, v)
```

The new point $\tilde{u}$ is the result of transportation of $u$ along vector $v$. To perform transportation of a vector along some other vector one can run the following line of code:

```
1 v_tilde = m.vector_transport(u, v, w)
```

Here we start from point $u$ and transport a tangent vector $v$ along a tangent vector $w$, and obtain $\tilde{v}$ that is the result of the vector transportation.

The last important method converts the Euclidean gradient of a function to the Riemannian gradient. Riemannian gradient is the search direction that takes into account the metric of a manifold and the tangent space in a given point. To calculate the Riemannian gradient one can use the following piece of code:

```
1 r = m.egrad_to_rgrad(u, e)
```

where we denote the Euclidean gradient as $e$ and the Riemannian gradient as $r$.

## 4.2 Optimizers

Riemannian optimizers implemented in QGOpt are inherited from TensorFlow optimizers and hence have the same API. The main difference is that one should also pass an example of manifold while defining an optimizer. An example of manifold guides the optimizer and preserves the manifold's constraints. Two optimizers are implemented, that are among the most popular in machine learning: Riemannian versions of Adam [18] and SGD [42].

If $m$ is a manifold element and lr is a learning rate, then the Adam and SGD optimizers can be initialized as follows:

```
1 # Riemannian ADAM optimizer
2 opt = qgo.optimizers.RAdam(m, lr)
3 # Riemannian SGD optimizer
4 opt = qgo.optimizers.RSGD(m, lr).
```

Note, that some other attributes, like value of momentum of SGD optimizer or AMSGrad modification of Adam optimizer, can be passed exactly as for TensorFlow Adam and SGD optimizers.

## 4.3    Auxiliary functions

It is important to have in mind that TensorFlow optimizers work well only with real variables. Therefore, one cannot use complex variables to represent a point on a manifold because they are being tuned while optimizing. The simplest way of representing a point from a complex manifold through real tensors is introducing an additional index that enumerates real and imaginary parts of a tensor. For example a complex-valued tensor of shape $(a, b, c)$ can be represented as a real-valued tensor of shape $(a, b, c, 2)$. During calculations, we need to convert tensors from the real representation to the complex representation and back. Let us assume that we initialize a complex-valued tensor, which represents a point from a manifold by using method "random". In order to make this tensor a variable suitable for an optimizer, one needs to convert it to the real representation. Then, while building a computational graph, one may need to have a complex form of a tensor again. To make this transition simple, we introduced two auxiliary functions that allow performing conversion from the real representation to the complex and back:

```
1 # a random real tensor, last index enumerates
2 # real and imaginary parts
3 w = tf.random.normal((4, 3, 2),
4                      dtype=tf.float64)
5 # corresponding complex tensor of shape (4, 3)
6 wc = qgo.manifolds.real_to_complex(w)
7 # corresponding real tensor (wr = w)
8 wr = qgo.manifolds.complex_to_real(wc)
```

# 5    Examples of application of QGOpt

## 5.1    Quantum gate decomposition

In this subsection we consider an illustrative example of a quantum gate decomposition. It is known, that any two qubit quantum gate $U$ can be decomposed in the following way [43]:

$$U = [\tilde{u}_{11} \otimes \tilde{u}_{12}]U_{\text{CNOT}}[\tilde{u}_{21} \otimes \tilde{u}_{22}] \times U_{\text{CNOT}}[\tilde{u}_{31} \otimes \tilde{u}_{32}]U_{\text{CNOT}}[\tilde{u}_{41} \otimes \tilde{u}_{42}], \qquad (7)$$

where $U_{\text{CNOT}}$ is the CNOT gate and $\{\tilde{u}_{ij}\}_{i,j=1}^{4,2}$ is a set of unknown one qubit gates. Since a set $\{\tilde{u}_{ij}\}_{i,j=1}^{4,2}$ can be seen as the direct product of 8 complex Stiefel manifolds, one can use Riemannian optimization methods to find all $\tilde{u}_{ij}$. First we initialize randomly a trial set $\{u_{ij}\}_{i,j=1}^{4,2}$ that will be tuned by Riemannian optimization methods. For the sake of simplicity let us denote the decomposition introduced above in the following way

$$D\left(u_{ij}\right) = [u_{11} \otimes u_{12}]U_{\text{CNOT}}[u_{21} \otimes u_{22}] \times U_{\text{CNOT}}[u_{31} \otimes u_{32}]U_{\text{CNOT}}[u_{41} \otimes u_{42}] \qquad (8)$$

The problem of gate decomposition can be formulated as the following optimization problem:

$$\|U - D(u_{ij})\|_F \to \min_{\{u_{ij}\}_{i,j=1}^{4,2}} \qquad (9)$$

where each $u_{ij}$ obeys the unitarity constraint and $\|\cdot\|_F$ is the Frobenius distance.

Before considering the main part of the code that solves the problem above, we need to introduce a function that calculates the Kronecker product of two matrices:

```
1  def kron(A, B):
2      AB = tf.tensordot(A, B, axes=0)
3      AB = tf.transpose(AB, (0, 2, 1, 3))
4      AB = tf.reshape(AB, (A.shape[0]*B.shape[0],
5                           A.shape[1]*B.shape[1]))
6      return AB.
```

Then we define an example of the complex Stiefel manifold:

```
1  m = qgo.manifolds.StiefelManifold().
```

As a target gate that we want to decompose, we use a randomly generated one:

```
1  U = m.random((4, 4), dtype=tf.complex128).
```

We initialize the initial set $\{u_{ij}\}_{i,j=1}^{4,2}$ randomly as a 4th rank tensor:

```
1  u = m.random((4, 2, 2, 2), dtype=tf.complex128).
```

The first two indices of this tensor enumerate a particular one-qubit gate, the last two indices are matrix indices of a gate. We turn this tensor into its real representation in order to make it suitable for an optimizer and wrap it up into the TensorFlow variable:

```
1  u = qgo.manifolds.complex_to_real(u)
2  u = tf.Variable(u).
```

We initialize the CNOT gate $U_{\mathrm{CNOT}}$ as follows:

```
1  cnot = tf.constant([[1, 0, 0, 0],
2                      [0, 1, 0, 0],
3                      [0, 0, 0, 1],
4                      [0, 0, 1, 0]],
5                      dtype=tf.complex128).
```

As the next step, we initialize Riemannian Adam optimizer:

```
1  lr = 0.2  # optimization step size
2  opt = qgo.optimizers.RAdam(m, lr),
```

and run the forward pass of computations:

```
1  with tf.GradientTape() as tape:
2      # turning u back into its
3      # complex representation
4      uc = qgo.manifolds.real_to_complex(u)
5      # decomposition
6      D = kron(uc[0, 0], uc[0, 1])
7      D = cnot @ D
8      D = kron(uc[1, 0], uc[1, 1]) @ D
9      D = cnot @ D
10     D = kron(uc[2, 0], uc[2, 1]) @ D
11     D = cnot @ D
12     D = kron(uc[3, 0], uc[3, 1]) @ D
13     # loss function
14     L = tf.linalg.norm(D - U) ** 2
15     # is equivalent to casting to a real dtype
16     L = tf.math.real(L).
```

The final step is to minimize the loss function $L = \|D(u_{ij}) - U\|_F^2$ that is calculated on the previous step. We calculate gradient of $L$ with respect to the set $\{u_{ij}\}_{i,j=1}^{4,2}$:

```
1  grad = tape.gradient(L, u),
```

and pass the gradient to the optimizer:

```
1  opt.apply_gradients(zip([grad], [u])).
```

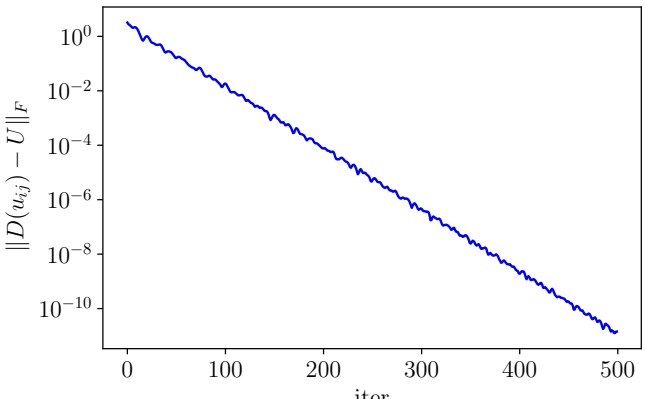

Figure 5: Frobenius distance between a gate and its decomposition. One can see that the distance rapidly decreases with the number of iteration towards almost a machine zero.

Adam optimizer performs one optimization step keeping orthogonality constraints. We repeat the forward pass, gradient calculation and optimization steps several times, wrapping them into a for loop until convergence and end up with a proper decomposition of the gate $U$. The optimization result is given in Fig. 5. One can see that at the end of the optimization process, the error is completely negligible. This section in the form of tutorial is available at the QGOpt documentation web-page [44].

## 5.2 Quantum tomography

Another typical problem that can be addressed by Riemannian optimization is the quantum tomography of states [45, 46] and channels [47, 48]. Here we consider an example of quantum tomography of channels, because it involves a more complicated structure than quantum tomography of states.

Let $\mathcal{H} = \bigotimes_{i=1}^{n} \mathbb{C}^2$ be the Hilbert space of a system consisting of $n$ qubits. Let us assume that one has a set of input states $\{\rho_i\}_{i=1}^{N}$, where $N$ is a total number of states, and each $\rho_i$ is a density matrix on $\mathcal{H}$. One passes initial states through an unknown quantum channel $\Phi_{\text{true}}$ and observes a set of measurement outcomes $\left\{ M_{k_i^1}^{\text{tetra}} \otimes \cdots \otimes M_{k_i^n}^{\text{tetra}} \right\}_{i=1}^{N}$, where $M_k^{\text{tetra}}$ is a tetrahedral POVM [49]:

$$M_k^{\text{tetra}} = \frac{1}{4} \left( \mathbb{1} + s_k^T \sigma \right), \ k \in (0, 1, 2, 3), \tag{10}$$

$$\sigma = (\sigma_x, \sigma_y, \sigma_z), s_0 = (0, 0, 1), s_1 = \left( \frac{2\sqrt{2}}{3}, 0, -\frac{1}{3} \right),$$

$$s_2 = \left( -\frac{\sqrt{2}}{3}, \sqrt{\frac{2}{3}}, -\frac{1}{3} \right), s_3 = \left( -\frac{\sqrt{2}}{3}, -\sqrt{\frac{2}{3}}, -\frac{1}{3} \right).$$

One can estimate an unknown channel by maximizing the logarithmic likelihood of measurement outcomes:

$$\sum_{i=1}^{N} \log \left( M_{k_i^1}^{\text{tetra}} \otimes \cdots \otimes M_{k_i^n}^{\text{tetra}} \Phi(\rho_i) \right) \to \max_{\Phi \text{ is CPTP}}. \tag{11}$$

For the sake of simplicity we assume that the many-body tetrahedral POVM $M$ is already predefined and has the shape $(2^{2n}, 2^n, 2^n)$, where the first index enumerates the POVM element. We also assume that we have a data set that consists of a set of initial density

matrices of shape $(N, 2^n, 2^n)$ and a set of POVM elements of the same shape that came true after measurements. In our experiments, an unknown channel has Kraus rank 2 and is generated randomly, initial density matrices are pure and also generated randomly.

Let us proceed with practical implementation. First, we define an example of the manifold of Choi matrices:

```
1  m = qgo.manifolds.ChoiMatrix().
```

The manifold of Choi matrices is represented through the quadratic parametrization with equivalence relation discussed in Section 3. Thus we initialize a variable, that represents the parametrization of a Choi matrix:

```
1  # random initial parametrization
2  A = m.random((2**(2*n), 2**(2*n)),
3  dtype=tf.complex128)
4  # variable should be real
5  # to make an optimizer work correctly
6  A = qgo.manifolds.complex_to_real(A)
7  # variable
8  A = tf.Variable(A).
```

Then we initialize the Riemannian Adam optimizer:

```
1  lr = 0.07
2  opt = qgo.optimizers.RAdam(m, lr),
```

and calculate the logarithmic likelihood function:

```
1  with tf.GradientTape() as tape:
2      # complex representation of parametrization
3      # shape=(2**2n, 2**2n)
4      Ac = qgo.manifolds.real_to_complex(A)
5
6      # reshape parametrization
7      # (2**2n, 2**2n) --> (2**n, 2**n, 2**2n)
8      Ac = tf.reshape(Ac, (2**n, 2**n, 2**(2*n)))
9
10     # Choi tensor (reshaped Choi matrix)
11     choi = tf.tensordot(Ac,
12                         tf.math.conj(Ac),
13                         [[2], [2]])
14
15     # turning Choi tensor to the
16     # corresponding quantum channel
17     phi = tf.transpose(choi, (1, 3, 0, 2))
18     phi = tf.reshape(phi, (2**(2*n), 2**(2*n)))
19
20     # reshape initial density
21     # matrices to vectors
22     rho_resh = tf.reshape(rho_in, (N, 2**(2*n)))
23
24     # passing density matrices
25     # through a quantum channel
26     rho_out = tf.tensordot(phi,
27                            rho_resh,
28                            [[1], [1]])
29     rho_out = tf.transpose(rho_out)
30     rho_out = tf.reshape(rho_out,
31                          (N, 2**n, 2**n))
32
33     # probabilities of measurement outcomes
34     # (povms is a set of POVM elements
35     # came true of shape (N, 2**n, 2**n))
36     p = tf.linalg.trace(povms @ rho_out)
```

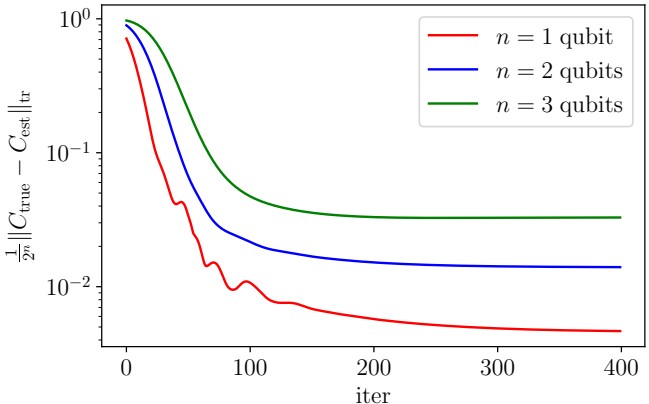

Figure 6: Dependence between Jamiołkowski process distance and number of iteration. Number of measurement outcomes $N = 600000$ for all experiments.

```
37
38      # negative log likelihood (to be minimized)
39      L = -tf.reduce_mean(tf.math.log(p)).
```

Finally, we calculate the logarithmic likelihood gradient with respect to the parametrization of the Choi matrix:

```
1 grad = tape.gradient(L, A),
```

and apply the optimizer that makes an optimization step that does not violate the CPTP constraints:

```
1 opt.apply_gradients(zip([grad], [A])).
```

We repeat the calculation of the logarithmic likelihood function, gradient calculation and optimization steps several times, wrapping them into a for loop, until convergence is reached. To evaluate the quality of an unknown quantum channel estimation we calculate Jamiołkowski process distance [50]:

$$J(\Phi_{\text{true}}, \Phi_{\text{est}}) = \frac{1}{2^n}\|C_{\text{true}} - C_{\text{est}}\|_{\text{tr}}, \tag{12}$$

where $\Phi_{\text{true}}(\Phi_{\text{est}})$ is the true (estimated) quantum channel, $C_{\text{true}}(C_{\text{est}})$ is the corresponding Choi matrix, $\|\cdot\|_{\text{tr}}$ is the trace norm and $0 \leq J(\Phi_{\text{true}}, \Phi_{\text{est}}) \leq 1$. One can see in Fig. 6 that the Jamiołkowski process distance converges to some small value with the number of iterations and we end up with a reasonable estimation of an unknown quantum channel. This section in the form of tutorial is available at QGOpt documentation web-page [44].

## 6 Discussion and concluding remarks

The range of applications of the QGOpt library to different problems of quantum technology is not limited only by quantum gate decomposition and quantum tomography. The six manifolds implemented in QGOpt give rise to different interesting scenarios of constrained optimization usage in quantum technology. For example, the complex Stiefel manifold can be used to address different control problems, where one needs to find an optimal set of unitary gates driving a quantum system to a desirable quantum state. Is is also possible to use a complex Stiefel manifold to perform entanglement renormalization [40, 41], machine learning by unitary tensor networks [51] or non-Markovian quantum dynamics identification [24]. Besides quantum tomography, quotient manifolds of density matrices

and Choi matrices can be used to maintain natural quantum constraints in different tensor network architectures. Quotient manifold of POVMs can be used for searching an optimal generalized measurement scheme, that gives maximal information gain. Finally, all these manifolds can be combined in one optimization task, which allows to address multi-component problems.

To conclude, we introduce QGOpt library that is aimed at solving constrained optimization problems with natural quantum constraints. We introduce and discuss quite an abstract concept, such as quotient manifolds, which lie under the hood of QGOpt. We go through QGOpt API and cover the most important features of it. We also sort out two examples of code solving two illustrative quantum technology problems.

## Acknowledgements

The authors thank Stephen Vintskevich and Mikhail Krechetov for fruitful discussions.

**Funding information** Authors are required to provide funding information, including relevant agencies and grant numbers with linked author's initials. Correctly-provided data will be linked to funders listed in the Fundref registry.

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
