# Peer review of "QGOpt: Riemannian optimization for quantum technologies"

_SciPost Physics Codebases_

## Round 1 · Referee Report · Anonymous (Referee 1) · 2020-11-24

Report
please find here enclosed my review of the manuscript entitled
QGOpt: Riemannian optimization for quantum technologies, submitted for publication in SciPost Physics.
The authors introduce a new optimization method for solving problems in constrained quantum dynamics. This approach is based on the Riemannian structure of the optimization problem which allows to apply a standard gradient algorithm, while preserving the physical constraints. Two examples in quantum technologies are discussed, namely gate decomposition and quantum tomography.
The development of efficient and versatile numerical optimization procedures is a subject of fundamental interest in quantum technologies both from a physical and mathematical point of view. The proposed method and the results are very interesting. The formulation is compact and seems straightforward to use. This paper seems sound mathematically and numerically. The different results are described in detail. I think that this paper is a significant contribution to this research area. It provides a complete and original investigation. I support the publication of this paper.
I have also minor comments about this manuscript. In the two examples presented by the authors, it would be interesting to describe briefly the computational cost of the algorithm. This information could be useful for the interested reader. In the same direction, is it possible to estimate the maximum dimension of the quantum system in which this approach can be applied? Another idea to discuss maybe in the conclusion of the paper: The authors use in this manuscript a first-order gradient algorithm. Would it be possible or interesting to extend this approach to second-order gradient algorithms? In the conclusion, the authors mention that "\emph{the complex Stiefel manifold can
be used to address different control problems, where one needs to find an optimal set of unitary gates driving a quantum system to a desirable quantum state}". In this perspective, would it be possible to combine this approach with optimal control theory? The authors should complete the bibliography of this paper about quantum optimal control. Quantum optimal control is for instance a very efficient procedure for generating quantum gates in a minimum time. They should cite a recent review paper (1) which gives the current state-of-the-art of optimal control in different domains, such as quantum information science. This review paper completes the other references mentioned in the current version of the manuscript.
(1)- S. J. Glaser et al., \emph{Training Schr\"odinger's cat: Quantum optimal control}, Eur. Phys. J. D (2015) 69, 279 [DOI: 10.1140/epjd/e2015-60464-1]
We thank the Referee 1 for the comments as well as the interesting and relevant paper suggested in the report. Our reply to the points raised follows.
1- In the updated version of the manuscript, we added the Appendix B that addresses the questions of complexity and scalability of algorithms. In particular, Table 3 shows the complexities of all optimization primitives for all the manifolds and can be used to estimate the complexity not only of the two examples from the text, but any other problem addressed by QGOpt. The answer to the question about a quantum system’s maximum dimension depends on a particular task and the particular approach. For example, in [A] we apply QGOpt to optimize Multi-scale Entanglement Renormalization Ansatz and find the ground state of the Transverse-field Ising spin chain consisting of 486 spins. Even though the Transverse-field Ising model is exactly solvable, this approach can be used for nonintegrable models. Therefore, for each particular problem and each particular approach, one needs to separately estimate the maximal dimension, which can be handled using Table 3.
2- One of the next steps of QGOpt's development is adding new types of optimizers. For example, the Quantum natural gradient descent [B] can be naturally generalized to the case of Riemannian embedded and quotient manifolds. Quasi-Newton methods like Riemannian BFGS [C] can also be integrated into the library, and more advanced first-order methods such as the Riemannian conjugate gradient method [D]. We added a paragraph to the last section of the updated version of the manuscript.
3- In the updated version of the manuscript, we cite the suggested paper [E] and a couple of other papers dedicated to optimal quantum control. Combining the optimization on the complex Stiefel manifold with optimal control theory looks like an interesting research question, which, however, deviates from the article’s topic. This question will be considered in further publications.
References
[A] Luchnikov I., Krechetov M., Filippov S. Riemannian optimization and automatic differentiation for complex quantum architectures arXiv preprint arXiv:2007.01287. – 2020.
[B] Stokes J., Izaac J., Killoran N., and Carleo G. (2020). Quantum natural gradient. Quantum, 4, 269.
[C] Huang W., Absil P. A. and Gallivan K. A. (2016). A Riemannian BFGS method for nonconvex optimization problems. In Numerical Mathematics and Advanced Applications ENUMATH 2015 (pp. 627-634). Springer, Cham.
[D] Edelman A., Arias T. A. and Smith S. T. (1998). The geometry of algorithms with orthogonality constraints. SIAM Journal on Matrix Analysis and Applications, 20(2), 303-353.
[E] Glaser S. J., Boscain U., Calarco T., Koch C. P., Köckenberger W., Kosloff R., et. al. (2015). Training Schrödinger’s cat: quantum optimal control. The European Physical Journal D, 69(12), 1-24.

Author: Henni Ouerdane on 2021-02-05 [id 1210]
(in reply to Report 3 by Michael Goerz on 2021-01-04)We thank Dr. Michael Goerz for the thorough and useful review. Our reply to the points raised in the report follows.
1- We revised the Introduction as suggested. We start from the statement that, in general, many quantum mechanics tasks can be formulated as optimization problems and list several examples, including the ground state search task, without linking with the Riemannian optimization. After that, we turn to the case of the optimization with constraints where QGOpt can be applied fruitfully.
2- In the updated version of the manuscript, we extend the beginning of section 2 and give a simple explanation of what we assume under the "transportation of points and vectors".
3- Since the language of tensor diagrams is one of the most compact and clearest ways of representing manipulations with tensors and since it becomes standard in fields of applied mathematics and quantum information sciences, we prefer to keep these diagrams in the manuscript.
In the updated version of the manuscript we make the discussion after Eq. (4) smoother and clarify the meaning of "isometric tensor." The simplest definition of the complex Stiefel manifold is a set of all isometric matrices of fixed size; we struggled to simplify it further but did not find the way. However we cite the following paper [A] in the text, that follows a "physicist’s style" of narration and provides all necessary information about the Stiefel manifold.
4- In the new version of the manuscript, we distinguish the quotient manifold ${\rm St}/Q$ (set of $A$ matrices) and Choi matrices $C_n$ but notice that they can be identified. The reason for that is that $C_n$ and ${\rm St}/Q$ are diffeomorphic as discussed in the new Appendix A. Therefore, optimization on ${\rm St}/Q$ is identical to optimization on $C_n$.
We apologize for the systematic mistake we made everywhere in the text. In the previous version, we used $A^\dagger A$ as a parametrization instead of the correct one $AA^\dagger$ that corresponds to the code of QGOpt. In the updated version of the manuscript we corrected this mistake. The index c of Ac indicates that Ac has a complex dtype and corresponds to $A$, not $A^\dagger$.
5- In the updated version of the manuscript, we introduce the new section 6 that shows the big picture. QGOpt supports solving problems defined on an arbitrary Cartesian product of manifolds (arbitrary concatenation of manifolds) implemented in QGOpt.
6- The use of "complex Stiefel manifold" in a quantum control problem ensures that $U$ is unitary. The case of a fixed number of piecewise-constant controls can be solved, for example by following the same approach as in the gate-decomposition example. It is definitely possible to approximate time-continuous control. Still, perhaps one can use the fact that each $U$ in this case is close to the identity matrix and simplify the overall procedure. In our opinion, it is an interesting research question that requires additional work. As mentioned in Section 6, QGOpt allows optimizing expressions that depend on multiple objects even from different manifolds.
7- The manifold of Hermitian positive definite matrices is introduced mostly for the completeness of the picture. We believe that it might be useful for some problems involving optimization over unnormalized density matrices. However, we do not have illustrative case in mind for now, but we are planning to find one and implement the solution of this task as a tutorial in the documentation. We have capitalized "Hermitian".
8- We have improved the quality of tutorials that already are in the documentation. We are going to add more tutorials as more instructive cases are found.
9- In the updated version of the manuscript, we introduce the new Appendix A that provides some mathematical details and cites sources of both types: for physicists and for mathematicians.
10- This notation is used in some papers and books in the optimization literature; see, e.g., [B].
11- In the updated version of the manuscript, we present a new Appendix B that includes a table with complexities of all optimization primitives and comparison of QGOpt with other libraries for Riemannian optimization.
References
[A] Edelman A., Arias T. A., and Smith S. T. (1998). The geometry of algorithms with orthogonality constraints. SIAM Journal on Matrix Analysis and Applications, 20(2), 303-353.
[B] Nesterov Y. (2003). Introductory lectures on convex optimization: A basic course (Vol. 87). Springer Science and Business Media.

---

## Round 1 · Referee Report · Anonymous (Referee 2) · 2020-12-15

Report
The manuscript presents QGOpt, a library for optimization based on a Riemannian point of view, adapted to optimization problems in quantum technologies. QGOpt is written on the top of TensorFlow. The manuscript is organized as follows: some general description of the principles of constrained Riemannian optimization is recalled, their role in quantum technology problems is discussed, and finally the library is described in its general lines and through two examples: quantum gate decomposition and quantum tomography. The presentation is rather enjoyable and the scope is clearly presented. However, no comparison with existing libraries is given, so it is difficult to asses its performances in comparison with the state of the art.
My background is not in the algorithmic aspects of the manuscript, so I will rather focus on the Riemannian part pointing to some issues with should, in my opinion, be improved before a possible acceptance.
-
it is odd to present Riemannian manifolds without mentioning geodesics at all and only marginally mentioning the inner product defining the Riemannian structure. Geodesics should at least be mentioned when speaking about retraction. The presentation of the Riemannian manifolds used in QGOpt (in particular, pp.6-7) should come with the choice of their inner products. For instance, in p. 5, in lines 6-7 from below, it is not clear how the set of complex isometric matrices is identified with a Riemannian manifold;
-
Riemannian gradient: at p.3 the authors speak of the "standard Euclidean gradient" of a function defined on a manifold M. If the function is not defined on a neighborhood of M, seen as an embedded submanifold of some Euclidean space, such an Euclidean gradient actually makes no sense;
-
in the brief presentation of optimization on Riemannian manifolds, it seems implicit that the considered manifolds are complete and without boundary. But the manifold \rho_n has boundary, hasn't it? And the manifold \mathbb{S}^n_{++} is maybe non-complete (it actually depends on the inner product, which is not specified). It seems to me that this issue should be raised;
-
p.8, the "orthogonal projection" to the tangent space is mentioned. As the authors know, orthogonal projection is meaningful only if a Riemannian structure is fixed. The ambient Euclidean structure, in the case of an embedded manifold, need not be an extension of the Riemannian structure, unless it is assumed that the Riemannian structure is the one induced by the embedding (and similarly for quotient manifolds of embedded manifolds, requiring some further metric properties of the quotient). If the Euclidean structure is not an extension of the Riemannian structure, then the orthogonal projection seems rather arbitrary and should be justified. All this should be clarified.
Minor remarks:
-
please do not use \mapsto in place of \to
-
I am not convinced that the diagrammatic representations in Figures 1,2,3 are helpful and pertinent to the discussion
-
p.5, first line after (5): "the set" instead of "a set"?
-
p.7, what do you mean by "essentially"?
We thank Referee 2 for the report where important remarks were made. Our reply to the points raised follows.
1- We introduce an exponential map through a geodesic in the updated version of the manuscript in section II. Only after this concept is introduced, we introduce the retraction map as a first-order approximation of the exponential map. However, we intend to retain the informal style of our presentation and keep it simple in the main text so that non-expert readers can gain a sound basis of how QGOpt works without having to delve deeply into mathematical details if they do not mean to. In the updated version of the manuscript, we introduce inner products for all the manifolds in Appendix A. Some of the manifolds in QGOpt support different types of inner product.
2- In practice, all functions are defined on a neighborhood of ${\mathcal M}$, and hence for embedded manifolds, one can introduce the Euclidean gradient. In Appendix A for a quotient manifold, we introduce the Riemannian gradient through the total manifold’s Riemannian gradient. A total manifold is an embedded manifold for which functions are defined on its neighborhood as well.
3- In QGOpt we use two types of inner product of $\mathbb{S}^n_{++}$. For both inner products the manifold is complete. We introduce both inner products in Appendix A. We also have a preprint [A] with a deeper discussion on optimization on $\mathbb{S}^n_{++}$ that we cite in the text. Regarding the manifold $\rho_n$ and other manifolds that are considered as quotient manifolds, we oversimplified the description and did not indicate that they are of a fixed rank. Under the condition of a fixed rank, they are manifolds without boundary. In the new version of the manuscript, we corrected this mistake in the text of section 3.
4- For embedded manifolds a Riemannian structure is induced by an ambient Euclidean structure for all inner products and thus the orthogonal projection is well defined. For quotient manifolds we use the orthogonal projection on the horizontal space of a total manifold (in our code the same method .proj as for embedded manifolds), which is also well-defined since the Riemannian structure of a total manifold is induced by the Euclidean structure of ambient space. The details are accessible for a reader in the Appendix A.
References
[A] Luchnikov I., Krechetov M., Filippov S. Riemannian optimization and automatic differentiation for complex quantum architectures. arXiv preprint arXiv:2007.01287. – 2020.

---

## Round 1 · Referee Report · Michael Goerz (Referee 3) · 2021-1-4

Strengths
1- High quality online documentation and code
2- Executable notebooks as tutorials
3- Overall accessible presentation of a mathematically abstract topic
Weaknesses
1- Unclear logical flow of introduction
2- Not all manifolds have examples in the online documentation
3- No automatic testing of package
4- No comparison / benchmarking comparing to existing packages, or discussion of numerical limitations.
Report
In their manuscript, the authors introduce a Python package "QGOpt" designed to solve certain optimization problems in quantum mechanics. The core idea of the package is to consider quantum objects such as unitaries, density matrices, or POVMs as elements of specific Riemanian manifolds that encode their inherent constraints. The package then implements two optimization methods that generally operate on Riemannian manifolds and thus will maintain those constraints. The manuscript provides an overview of how this optimization works conceptually, explains the interpretation of quantum channels as a "quotient Stiefel manifold", presents the core API of the QGOpt package, and provides two example usages, optimal quantum gate composition and quantum channel tomography. The online documentation of the package additionally contains examples for "Entanglement normalization", quantum state tomography, and POVM optimization. In line with the review guidelines of SciPost Physics Codebases, my comments here, including the list of Strengths/Weaknesses extend over the submitted manuscript, as well was the package code published on Github and the online documentation.
Overall, the manuscript presents an intriguing approach to solving certain optimization problems in quantum mechanics. The author convincingly motivate their approach, making the QGOpt package a welcome addition to the open source ecosystem for quantum technology. Both the manuscript and the underlying package are of high quality, and I fully support publication. However, I have some suggestions to improve the clarity of the manuscript, as well as some concerns about completeness of the online documentation, which I'll detail in the following (numbered for easier response). Most of these are optional and in addition to the "requested changes".
1- I found it difficult to follow the logical structure of the introduction. It seems to me like the Introduction should start with the third paragraph ("Some problems of quantum mechanics require nonstandard optimization method [...]"). The first and second paragraphs then seem like specific examples (ground state identification and quantum tomography) that should follow the general problem statement. I was especially confused by the first example of ground state identification, as it is not entirely clear whether QGOpt is intended as a solution to this specific problem. Is it? Is there a manifold corresponding to pure Hilbert space states, or does one need to elevate Eq (1) to density matrices?
2- Section 2 of the manuscript, giving an overview of the Riemannian optimization is particularly well-written, providing an intuitive and easy to understand discussion of a mathematically abstract topic. It might be good to slightly expand the first sentence of this section, though: the change of perspective from a straightforward evaluation of objectives in Euclidean space to "points and vectors transportation" which then leads to the idea of Riemannian optimization is certainly not something the average physicist is used to. It would be good to illuminate this with a brief example: in some trivial evaluation of a functional in Euclidian space, what exactly is the "transportation"? Meanwhile, I found the description of the "retraction" and the steps 1-3 of the Riemannian generalization of gradient descent very understandable with my limited knowledge of Riemannian manifold (basically just something that looks like a Euclidian space in a sufficiently small region, but "curved" or "constrained" when seen from the large space it is embedded in, with the canonical example of the surface of a globe in three-dimensional space).
3- Section 3, in comparison, does not quite manage to address quite as well the standard physicist with that same basic understanding of Riemannian manifolds. I'm not sure how much the foray into the diagrammatic representation contributed (not that it hurts, but if pressed for brevity, it could probably be cut). More importantly, though, the discussion becomes a bit abrupt after Eq (4). I have no idea what an "isometric tensor" is (presumably a tensor that preserves a metric; the Wikipedia entry on isometry was not really that helpful). It would be good to give an intuitive definition of "isometric tensor" in this context and to explain why Eq (5) implies that $A_{resh}$ is the set of isometric tensors. Likewise, is there some intuitive "physicist's definition" of what a Stiefel manifold is?
4- In the bullet list of all the manifolds on page 6, the "Quotient manifold of Choi matrices" is the one that is discussed in the main part of section 3, right? But, it's the A that is the manifold (does A have a name?), not the C, right? In an optimization, one has to optimize for A and then get the corresponding C from Eq. (4) in the example code listing on page 12, is the variable "Ac" $A^\dagger$? This would then indeed make line 11 into $A^\dagger A$ corresponding to Eq (4). Is there a reason to work with $A^\dagger$ is a variable instead of $A$?
5- Either before or after the complete lists of manifolds supported by QGOpt it might be good to paint the big picture. As I understand it, QGOpt allows be to solve the general optimization problem
$$\arg\min_A f(A)$$
where A is an element of one of the supported manifolds and f is an arbitrary function that can be evaluated within the tensorflow framework, cf. Eq (1) (which brings me back to my earlier question: which manifold is $\Psi$ in?)
It might be good to write this out explicitly.
6- I'm also particularly interested in the manifold of the unitary matrices and the application of quantum control (the first bullet point). Suppose I have a state-to-state optimization problem. In terms of the above control equation, this would be formulated as
where the use of the "complex Stiefel manifold" in QGOpt ensures that U is in fact unitary, correct? Most directly this means I could solve the control problem for one single time-independent control Hamiltonian (the matrix-log of U). Would it be conceivable to extend this to slightly more elaborate use cases, e.g. a fixed number of piecewise-constant controls, that is, a set of U's instead of a single U? Do I follow the same approach as in the gate-decomposition example (concatenating the unitaries)? How for could I push this? Would it still be numerically feasible to approximate time-continuous control, e.g. with a thousand time steps (a thousand unitaries)? In general, could QGOpt allow to optimize expressions that depend on multiple objects even from different manifolds (not that I have a specific example in mind)?
7- It might be good to elaborate on the manifold of "Hermitian positive definite matrices" (last-but-one in the bulleted list; also: capitalize "Hermitian"). In which context do these occur?
8- In fact, I would recommend that the online documentation of QGOpt should contain at least one example for each of the supported manifolds.
9- For each of the manifolds, is there a citable source that explains why these mathematical objects can be viewed as elements of those manifolds? These are clearly not necessary for usage of the package, but it would still be good to have some references for anyone who wants to dive deeper into the topic. From this perspective, more didactic sources "for physicists" would be helpful, although the mathematical pendants might also ask for "mathematical proofs".
10- The notation with the arrow in Eq (9) is a bit weird.
11- There is no discussion of limitations, e.g. how the optimization might scale numerically. This somewhat depends on the answer to my question of whether it is possible to optimize for more than one object at the same time (if not, the dimension of the control problem is likely always "small"). It might be good to benchmark against other methods or software packages to the extent that they are available. From my perspective, this is very much optional, although the review guidelines do ask about "Benchmarking tests must be provided."
Requested changes
The following requested changes are primarily within the software package/documentation on Github, less so in the manuscript. I will also add Github issues for some of these.
1- Specify a package version to which the manuscript pertains. I noticed there has been unreleased development on the master branch. If you want to follow "semantic versioning", you may want to declare the package API "stable", release a 1.0, and write the manuscript specifically for that.
2- The package needs to have automatic testing. The example notebooks are very nice, but not automatic (ensuring that future development will not break the package, or make it easy for new contributors to add code). I noticed that the repository includes some test code. I would recommend to set up continuous integration (Github Actions) to run the tests automatically. Also, the documentation must include some guidelines for contributors, most importantly how to run the tests manually.
3- Consider adding tutorials for all the supported manifolds. This is somewhat optional, and could be done after publication. On the other hand, it might uncover bugs/problems that you'd want to fix before publication or a 1.0 release. Rename "Quick Start" to "Quick Start: Quantum Gate decomposition"
4- Check the capitalization of titles in references, e.g. "Monte Carlo" and other proper nouns should always be capitalized.
5- There are some minor language problems (missing articles, punctuation). Consider having a native speaker read through the manuscript.

---

## Editorial Decision

unknown